# Role of Mesenchymal Stem Cells and Their Paracrine Mediators in Macrophage Polarization: An Approach to Reduce Inflammation in Osteoarthritis

**DOI:** 10.3390/ijms232113016

**Published:** 2022-10-27

**Authors:** Sree Samanvitha Kuppa, Hyung Keun Kim, Ju Yeon Kang, Seok Cheol Lee, Jong Keun Seon

**Affiliations:** 1Department of Biomedical Sciences, Chonnam National University Medical School, Hwasun 58128, Korea; 2Department of Orthopaedics Surgery, Center for Joint Disease of Chonnam National University Hwasun Hospital, 322 Seoyang-ro, Hwasun-eup 519-763, Korea; 3Korea Biomedical Materials and Devices Innovation Research Center, Chonnam National University Hospital, 42 Jebong-ro, Dong-gu, Gwangju 501-757, Korea

**Keywords:** osteoarthritis, macrophage, inflammation, macrophage polarization, mesenchymal stem cells

## Abstract

Osteoarthritis (OA) is a low-grade inflammatory disorder of the joints that causes deterioration of the cartilage, bone remodeling, formation of osteophytes, meniscal damage, and synovial inflammation (synovitis). The synovium is the primary site of inflammation in OA and is frequently characterized by hyperplasia of the synovial lining and infiltration of inflammatory cells, primarily macrophages. Macrophages play a crucial role in the early inflammatory response through the production of several inflammatory cytokines, chemokines, growth factors, and proteinases. These pro-inflammatory mediators are activators of numerous signaling pathways that trigger other cytokines to further recruit more macrophages to the joint, ultimately leading to pain and disease progression. Very few therapeutic alternatives are available for treating inflammation in OA due to the condition’s low self-healing capacity and the lack of clear diagnostic biomarkers. In this review, we opted to explore the immunomodulatory properties of mesenchymal stem cells (MSCs) and their paracrine mediators-dependent as a therapeutic intervention for OA, with a primary focus on the practicality of polarizing macrophages as suppression of M1 macrophages and enhancement of M2 macrophages can significantly reduce OA symptoms.

## 1. Introduction

Osteoarthritis (OA) is the most common chronic musculoskeletal condition that affects weight-bearing joints, such as the spine, hips, knees, and hands [1]. OA affects approximately 300 million people globally [2] and is ranked sixth in Asia and eleventh worldwide in terms of years lived with disability (YLD) [3]. Stiffness, decreased range of motion, joint instability, edema, muscle weakness, weariness, and pain-related psychological distress are the most typical symptoms of OA. The risk factors involved in OA interact in a complex manner; they can be divided into individual-level risk factors (age, gender, obesity, heredity, and diet) [4] and joint-level risk factors (injuries, misalignments, and incorrect joint loading) [5]. OA has significant economic effects due to the growing frequency of joint replacements, increasing medical costs, and an aging population [6]. As the population ages, OA is expected to become the primary cause of disability in elderly people by 2030 [7].

OA was initially thought to be caused by wear and tear of the joint’s cartilage and bony understructure due to constant mechanical stress. However, the pathogenesis of OA is remarkably more complicated. Instead of a simple wear-and-tear condition, OA is now recognized as a complex ailment in which inflammation plays a significant role in joint deterioration [8]. The synovium is the principal site of inflammation in OA. Macrophages have a major role in the inflammation of the synovium in OA. Synovial macrophages promote the progression of OA by initiating an inflammatory response through the production of pro-inflammatory cytokines and mediators that control immune system activity and encourage the secretion of additional pro-inflammatory mediators by cartilage and synovial cells [9]. Numerous researchers have demonstrated that inhibiting inflammation either by reprogramming macrophages or depleting them has proven to be effective [10]. The primary reason for targeting macrophages is their remarkable plasticity and capacity to fulfill several biological roles in response to signals from the tissue microenvironment. Numerous strategies have been developed to target these properties; however, no macrophage-specific treatment is currently available on the market for any condition.

Mesenchymal stem cells (MSCs) play an essential role in the polarization of macrophages [11]. There has been a significant shift in perspective on the role of MSCs in recent years. MSCs were first thought to be therapeutically useful because of their potential to differentiate into several other cell types that might potentially replace cells in damaged or diseased tissues [12]. It is now well accepted that MSCs’ paracrine secretion is responsible for the vast majority of their therapeutic effects. The paracrine secretions consist of soluble proteins, free nucleic acids, lipids, and extracellular vesicles (EVs) collectively known as the secretome. The secretome has been shown to promote remodeling of the extracellular matrix (ECM), manage local inflammation, and enhance macrophage polarization by decreasing the release of pro-inflammatory factors and increasing anti-inflammatory factors [13,14,15].

In this article, we review the current understanding of the role of macrophages in OA inflammation and the existing techniques to polarize macrophages. We will also explore the recent developments in the use of MSC secretome to polarize macrophages in diverse inflammatory models that will help in facilitating the translation of future research for the treatment of inflammation in OA.

## 2. Inflammation in Osteoarthritis

OA has always been an uncertain condition in terms of inflammation even though the name itself denotes that it is an inflammatory process. Previously it was believed that OA was caused due to biomechanical causes and was employed as a negative control for inflammation during comparisons to rheumatoid arthritis (RA) [16]. It has now been brought to light that OA is much more than just an injury caused due to overuse of the joint. OA is a complex biological response as a result of its interaction with tissue resident cells and their mediators which amplifies physical stress incapacitating the normal function of ligaments, muscles and menisci [17,18]. Research has identified the process of inflammation as the initial step along the negative chain of events that leads to early OA.

Although the inflammatory response in OA is not as pronounced as RA, several authors have confirmed there is low-grade inflammation in OA [19]. The presence of inflammation in OA has been studied extensively using various techniques, in the early 1980′s Goldenberg et al. exhibited that majority of the inflammation is present in the synovium of the OA patients through histopathological analysis [20]. Additionally, the histological evidence was validated when correlated to the levels of serum C-reactive protein (CRP) and the levels of the inflammatory marker interleukin-6 (IL-6) in the synovial fluid in patients undergoing total hip or knee arthroplasty [21]. A comparative study between synovial tissues of patients with early and late OA revealed increased infiltration of mononuclear cells and inflammatory cytokines in patients with early knee OA to late knee OA [22]. Later, due to sensitive imaging techniques like magnetic resonance imaging (MRI) and direct arthroscopic visualization, it was confirmed that inflammation is visible in the synovium at the early stages of OA even before there is visible articular cartilage damage [23]. A recent study revealed that synovitis is one of the key factors in identifying early OA which was confirmed through the analysis of serum matrix metalloproteinase-3 (MMP-3) concentration, effusion-synovitis volume and synovial score [24]. All these studies validate the significant role of synovitis at any (early or late) stage of OA.

In the past decade, researchers have also been investigating the connection between low-grade synovitis and the manifestation of OA’s clinical symptoms. Synovitis has been linked to more severe symptoms including pain and joint dysfunction and may generate a more rapid deterioration of cartilage [4]. Synovitis has been linked to symptoms such as discomfort in people with knee OA. The correlation between pain and synovitis on MRI, found that changes in pain levels over time corresponded with changes in synovitis, lending credence to the idea that the two are causally connected [25]. Recently, a similar relationship between pain and synovitis was described using contrast-enhanced MRI wherein the likelihood of experiencing painful knee OA was found to rise 9-fold with increasing synovitis severity [26]. Ayral et al., published a study that established a link between synovitis and the progression of cartilage degeneration. The presence or absence of synovitis and the overall health of the cartilage surfaces were easily discerned during the initial arthroscopy. The rate of cartilage degeneration was measured by an arthroscopic examination performed 12 months after the original surgery. The presence of synovitis was related with more severe chondropathy at baseline and was present in around 50% of patients. Moreover, at one year, patients with synovitis were more likely to have advanced cartilage pathology than those without the inflammation [27].

This suggests that synovitis is associated with pain and cartilage erosive mechanisms, making it a possible target for disease- and symptom-modifying therapy. The inactivation of critical inflammatory pathways either by removing macrophages or reprogramming them can reduce joint structural pathology, including cartilage degradation. Although inflammation does not pertain to synovial tissue alone, synovium is the major site of gross and microscopic inflammatory change [28], and targeting inflammation can further reduce the vicious cycle of the disease, thus making it the major focus of this review.

## 3. Inflammatory Mediators Secreted by Macrophage and Its Interaction with Resident Cells during OA

During OA, the macrophages fail to keep up their stability and are activated through various ways mainly when macrophages are stimulated by damage associated molecular patterns (DAMPs) or pathogen-associated molecular patterns (PAMPs) upon interaction with germline-encoded surface pattern recognition receptors (PRRs) on macrophages; they activate the nuclear factor kappa-light-chain-enhancer of activated B cells (NF-κB) pathway, causing the cells to release an increased amount of inflammatory mediators [29]. Another key signaling channel is the NOD-like receptor family, pyrin domain-containing 3 (NLRP3) inflammasome-mediated pathway. Both pathways can activate the macrophages during OA and trigger the production of two of the most extensively studied pro-inflammatory cytokines, interleukin-1β (IL-1β) and tumor necrosis factor (TNF-α). The cascade of mediators activated by the pro-inflammatory cytokines is shown in Figure 1.

IL-1β released by macrophages stimulates chondrocytes to synthesize MMPs, especially collagenase-1 (MMP-1), stromelysin (MMP-3), collagenase-3 (MMP-13), and ADAMTS-4 and 5, which are known to cause cartilage degradation and synovial damage [30,31]. IL-1β increases the production of other cytokines, such as IL-6, and IL-8 and chemokines, such as CCL2 (monocyte chemoattractant protein-1, MCP-1), macrophage inflammatory protein-1 alpha (MIP-1α/CCL3), and C–C motif chemokine ligand 5 (CCL5) in chondrocytes via a paracrine mechanism [32]. These mediators attract new macrophages to the joint, where they continue to release IL-1β, thereby prolonging the inflammatory cycle [33]. Additionally, IL-1β promotes the release of a variety of pro-inflammatory mediators, such as prostaglandin E2 (PGE2), nitric oxide (NO), and cyclooxygenase-2 (COX-2), which stimulate the extracellular signal-regulated kinases (ERKs) pathway. Activation of the ERK pathway inhibits type II collagen and aggrecan formation, as well as ECM synthesis [34]. Likewise, the c-Jun N-terminal kinases (JNKs) pathway also inhibits collagen II synthesis by inhibiting the SOX-9 gene. The mitogen-activated protein kinase (MAPK) signaling cascades regulates MMP-1, MMP-13, and ADAMTS-4, whereas MMP-3 and ADAMTS-5 are exclusively regulated by the ERK and JNK pathways, respectively [35].

Similarly, TNF-α also exhibits comparable effects on chondrocytes, increases IL-6, IL-8 and IL-18, suppresses the production of proteoglycans and type II collagen, and stimulates chondrocytes to generate MMPs and ADAMTS for ECM degradation [36,37]. Alternate to secreting inflammatory mediators, activated macrophages also produce growth factors. Vascular endothelial growth factor (VEGF) contributes to the severity and inflammation of OA. The articular cartilage, serum, and synovium of patients with late-stage OA show elevated VEGF expression. An increase in angiogenesis and VEGF production is the root cause of synovitis. VEGF has been shown to increase symptoms in patients with OA by stimulating inflammation-promoting macrophages to migrate throughout the inflamed tissue while also delivering nutrition and oxygen [38]. Other growth factors, including bone morphogenetic protein-2 (BMP-2), bone morphogenetic protein-7 (BMP-7), and transforming growth factor beta (TGF-β), contribute to the production of osteophytes and synovial fibrosis. Apart from osteophyte production, macrophages induce the formation of osteoclasts, which disintegrate the underlying bone and further degrade the cartilage and bone beneath by prompting osteoblasts to release a variety of cytokines and MMPs [39]. In addition to chondrocytes, activated macrophages stimulate neighboring fibroblast like synoviocyte (FLS) to produce a variety of inflammatory cytokines and chemokines, as well as pro-inflammatory mediators resulting in synovial hyperplasia, joint swelling, inflammation, and pain. FLS secrete a clear, viscid, lubricating fluid known as synovial fluid [40]. Macrophages also impair the primary role of FLS of maintaining cartilage homeostasis and shielding the cartilage surface from friction and deterioration [41]. The role of macrophages in inflammation is owed to their secretion of inflammatory cytokines. The majority of the cells within the joint interact with the cytokines released by macrophages (as shown in Figure 1), and these interactions influence the production of cytokines, other inflammatory mediators, and enzymes by these cells via intracellular signal transduction pathways, which plays a crucial role in the pathogenesis of OA.

## 4. Depletion of Macrophage

Macrophages’ role in inflammation emphasizes that they have a critical role in OA immunopathogenesis and are not just a consequence of it. Therefore, depletion of macrophages can be a potential intervention that promote tissue repair and remodeling. Blom et al. demonstrated that removal of macrophages from the synovial lining decreased the expression of MMP-3 and MMP-9 in the synovium but not in the cartilage, and also reduced osteophyte formation in the collagenase induced OA (CIOA) mouse model [42]. Bondeson et al., showed that depleting synovial macrophages with anti-CD14-conjugated magnetic beads reduced inflammatory cytokines [43]. Correspondingly, the pro-inflammatory cytokines produced by FLS cease to secrete these cytokines and halts ECM degradation [44]. Wu et al. sought to determine whether the same is true for obesity-related OA and discovered that short-term macrophage depletion elevated synovitis and T-cell and neutrophil infiltration into the operated joint. These researchers concluded that macrophages are essential regulators of the responses of other immune cells and macrophage depletion cannot be employed to reduce inflammation in obese arthritic patients [45]. However, these results oppose those of Sun et al., who demonstrated that clodronate-liposome-mediated macrophage depletion and resolution of inflammation using a pro-resolving lipid mediator, resolvin D1, reduce pro-inflammatory gene expression and enhance anti-inflammatory gene expression in a similar obesity-associated mouse model of OA [46].

Crucial components to consider include the number of rounds of macrophage depletion and the type of depletion, which explains the reduction in OA severity observed in the clodronate-liposome-mediated depletion model, which was subjected to local depletion and frequent injections [46]. Although macrophage depletion is effective in reducing the severity of OA and inflammation, there are a few disadvantages associated with this approach. Macrophages are not only agents of destruction, but also play critical defensive and reparative roles in the host. Therefore, their depletion may have unintended consequences. When inflammation is localized to a single organ, systemic depletion of macrophages will have a major impact on the ability of macrophages to maintain homeostasis in all healthy tissues, which is clearly not a promising therapeutic approach [47]. Finally, macrophage depletion impairs the host immune system, which should be avoided. 

## 5. Macrophage Phenotype and Polarization

Often, during an inflammatory response, macrophages can exhibit a spectrum of phenotypes; however, the two most frequent phenotypes that define macrophages are classically stimulated M1 and alternatively stimulated M2 macrophages [48]. M1 macrophages (CD80^+^, CD86^+^) mainly exert pro-inflammatory effects; they are formed owing to numerous stimuli, such as TNF-α, interferon gamma (IFN-γ), or lipopolysaccharide (LPS), leading to the release of large amounts of pro-inflammatory cytokines (IL-1β, IL-6, IL-8, and IL-18) [49]. M2 macrophage phenotype (CD163^+^ and CD206^+^) pertains to tissue repair and downregulation of inflammation and secrete anti-inflammatory cytokines (IL-4, IL-10, and IL-13) [49]. The presence of both M1 and M2 macrophages in the synovium has been described by Liu et al.; they conducted a study which analyzed the ratio of M1/M2 macrophages in human normal vs. OA knee samples and concluded that the increase in M1/M2 ratio positively corresponded to the severity of OA classified through the level of Kellgren Lawrence grade of OA in the knee [50]. Furthermore, the ratio of M1/M2 was also studied in canine [51] and equine models [52] wherein the synovial fluid samples showed higher M1/M2 ratio compared to normal groups. The presence of inflammatory macrophages in equine model was also confirmed through coculture of osteochondral-synovial explant ex vivo OA model wherein the ratio of NO (µM)/urea (µM) increased over time, suggesting that macrophages in the synovium gradually underwent a shift to M1 phenotype [53]. Another study correlated the radiographic OA intensity and symptoms with the quantity of activated macrophages present in the OA knee joints detected utilizing the imaging agent 99mTc-EC20 (etarfolatide), which specifically binds to folate receptor β (FR β) on activated, but not resting, macrophages or other immune cells [54]. Zhang et al. showed that predominantly M1 macrophages accumulated in human and mouse OA synovial tissue and not M2 macrophages. Further confirmation was provided with the help of a transgenic mouse model having enhanced M1 or M2 macrophages, the M1 macrophages in the synovium aggravated CIOA whereas the presence of M2 macrophage downregulated the development of OA [55]. Studies carried out with the help of anterior cruciate ligament transection (ACLT) rodent model and destabilization of medial meniscus (DMM) murine model, two well established animal models of OA also reported higher number of F4/80^+^ CD86^+^ and nitric oxide synthase 2 (NOS2^+^) M1 macrophages in the synovium [55]. However, data collected using the *in-silico* method CIBERSORT using publicly accessible transcriptome information revealed an abundance of M2 macrophages (30.1%), resting T-cells (23.9%), and activated NK cells (16.2%) in the synovial tissue of OA patients. While these statistics differ somewhat from the immunological profile of normal synovium (26.8%, 24.1%, 15.0%, respectively), the increase in M2 macrophages was statistically significant [56]. However, further research is needed to determine the importance of these alterations and the processes by which they may develop.

Currently, the options for treating OA are very limited. The already existing conventional therapeutic approaches such as physiotherapy, pharmacological drugs and surgery are not adequate as they are not significant enough to modify the prevailing course of the disease or prevent the process of cartilage degeneration. There is a significant need for disease modifying therapeutic intervention for OA. Recently, researchers have focused their attention on targeting macrophages due to their high plasticity and ability to perform distinct biological functions based on the signals received within the tissue microenvironment. In spite of the hypothesized detrimental role of activated macrophages in OA, their systemic depletion was found to be fatal. Instead, reprogramming macrophages may be a future therapeutic strategy [57]. Polarizing macrophages to an anti-inflammatory phenotype holds great promise for the treatment of inflammation in OA. Currently many biomolecules derived from herbal plant extracts, nanoparticles and neutralizing antibodies are gaining a lot of attention for their macrophage polarization ability owing to their anti-inflammatory properties and immunoregulatory activities. Here we summarize the different molecules currently available and the mechanism by which they polarize macrophages (Table 1).

## 6. Macrophage Polarization by Mesenchymal Stem Cells

All of the aforementioned ways for polarizing macrophages are capable of reducing inflammation and pain, but they cannot repair the cartilage. Further cartilage deterioration requires complete knee arthroplasty [72,73]. OA treatment has the potential to be revolutionized by stem cell treatment. MSCs have piqued the interest of many researchers because of their experimental applicability, and the ability to differentiate into many lineages such as bone, muscle, fat, and cartilage [12]. MSCs can be extracted from bone marrow, synovium, adipose tissue, umbilical cord, blood, dental pulp and endometrium [74]. The International Society for Cellular Therapy (ISCT) has established three baseline requirements that all MSCs, regardless of their origin, must satisfy. Initial attachment to the plastic surface is required for MSCs to proliferate under standard circumstances. Additional requirements for MSCs include the expression of the surface markers CD73, CD90, and CD105. Furthermore, MSCs are required to differentiate into osteoblasts, chondrocytes, and adipocytes under certain in vitro conditions [75]. One of the most compelling arguments for making MSCs a standard treatment for OA is that they can repair cartilage, allowing damaged cartilage to regenerate [76]. MSCs are self-renewing stromal cells that can develop into a variety of cell types [77]. Although Friedenstein was the first to effectively isolate bone-forming cells from a guinea pig, Owen provided this field of research a much-needed boost by extending it to rats [78]. In 1992, it was announced that human bone marrow MSCs (BM-MSCs) had been extracted and cultivated to increase in number; by 1995, they were being pumped into patients [79]. Over the past quarter century, infusion approaches have demonstrated such a high level of safety that the Food and Drug Administration (FDA) now lists more than 950 clinical trials involving MSCs. MSCs have been used to treat a variety of orthopedic disorders, including OA, due to their tissue regeneration and immunomodulatory properties. MSCs have been shown to be effective in treating OA in phase 1 trials over the last few years and a number of unpublished Phase 2 trials, notably ADIPOA2 [80,81].

MSCs have an effect on immune cells, such as macrophages, dendritic cells, T lymphocytes, and natural killer (NK) cells [82]. MSCs play a pivotal role in controlling different functions of macrophages, such as differentiation of naive macrophages, modulation of their phagocytic ability, enhancement of their bactericidal effect, and manipulation of the plasticity and polarity of macrophages. MSCs have been reported to possess the property of immune evasiveness due to their close and reciprocal interaction with immune cells and their immunomodulatory properties [83]. This suggests that MSCs may be immune-protected when injected into an allogenic environment, preventing detection and rejection by the immune system. Nevertheless, MSCs do not need to remain in the body for an extended period of time in order to exert a therapeutic effect. A brief presence can permanently alter tissue cell behavior under certain pathological conditions. Thus, it is necessary to comprehend how the host immune system reacts to allogenic MSCs and how this may influence the therapeutic efficacy of MSCs in various inflammation models. MSC-based therapies have been developed in the context of inflammation observed in numerous disease models, such as graft versus host disease (GVHD) [84], inflammatory bowel disease (IBD) [85], diabetic cardiomyopathy [86], and many others. In this review, we will discuss some of the fundamentals that link inflammation to different disease models, as well as some of the biological properties of MSCs that can help them be used as a treatment for inflammatory diseases like OA. MSCs manifest their immunomodulatory properties and induce polarization of macrophages to an anti-inflammatory M2 phenotype through two primary methods: cellular interaction and paracrine factor-mediated mechanisms involving cytokines and hormones, and exosome-mediated mechanisms involving RNAs, and other molecules as shown in Figure 2.

## 7. Cellular Interaction

### 7.1. Immunosuppression through MSCs-Macrophage Interaction

Recent studies have revealed the ability of MSCs to regulate macrophage polarization through direct cellular contact. Abumaree et al. reported human placental MSCs (hp-MSC) mediated differentiation of macrophages from M1 into M2 either through direct cellular contact or through partial interaction of various soluble mediators such as macrophage colony-stimulating factor (M-CSF), IL-10, B7-H4, Leukemia inhibitory factor (LIF), VEGF, PGE2, TGFβ-1 with glucocorticoid receptor and progesterone receptor [87]. The results of Yanhong Li et al. demonstrate that direct cell-to-cell contact between MSCs, and pro-inflammatory macrophages is significant. The crosstalk between MSCs and macrophages was enhanced when they were in direct contact with each other due to the stimulation of macrophages that upregulated the expression of CD200 in MSCs and M1 macrophages expressed the CD200R receptor, which enabled the contact, and this linkage drove the M1 to M2 transition [88]. Audrey Varin et al. observed a similar interaction between M1 macrophages and MSCs, in which the interaction was with the accumulated CD54 marker generated in the interface of MSC-M1 macrophages that induced Ca^2+^ ion signaling and enhanced MSC’s immunosuppressive ability [89]. In response to inflammation, MSCs upregulate the expression of CD54, one of their most abundant adhesion molecules [90]. In multiple studies, CD54-Leukocyte function associated antigen-1 (LFA-1) interactions have been shown to determine CD54′s ability to polarize macrophages. Takizawa et al. have shown that CD54-LFA1-mediated interaction between MSCs and macrophages in hypoxia which prompted the proliferation of M2-macrophages in mice [91]. Additionally, CD54^high^ MSCs enhance mouse survival in a GVHD model by migrating to secondary lymph nodes to inhibit dendritic cell maturation and Th1 differentiation of CD4^+^ T-cells [92].

### 7.2. Immunosuppression through Efferocytosis

Apoptotic cells communicate with immune cells either directly or indirectly via phagocytosis. In the direct method, the immune cells directly interact with the apoptotic cells, creating an immunosuppressive environment by releasing IL-10 and TGF-β to dampen the pro-inflammatory response of LPS-stimulated macrophages that secrete IL-1β and TNF-α [93,94]. Indirect effects reduce LPS reactivity, through phagocytosis of apoptotic cells which reduces the inflammatory phenotype of the immune system. Galleu et al. demonstrated that in vivo naturally occurring MSC apoptosis is instrumental for the delivery of immunosuppression. In a model of GVHD, the release of granules containing the perforin granzyme B by recipient T-cells caused the apoptosis of MSCs. The regulatory T-cells and macrophages ingest apoptotic MSCs and release indoleamine 2,3-dioxygenase (IDO), restoring an anti-immunogenic environment [95]. Similarly, Akiyama et al. found that MSCs can also promote an anti-inflammatory milieu by controlling T-cell apoptosis via the FAS/FASL pathway, causing macrophages to produce large amounts of TGF-β, which then upregulates regulatory T-cells, resulting in immunological tolerance [96]. These findings provide credence to the theory that MSCs can provide therapeutic benefit without engraftment. As stated previously, these results demonstrate that despite their brief post-administration survival, MSCs induce potent immunosuppression.

Although cellular interaction is an essential functional mechanism to regulate macrophage polarization, recent studies indicate that the immunomodulatory properties of MSC are largely dependent on the paracrine mediators secreted by MSCs.

## 8. Paracrine Mediators

MSCs secrete a plethora of paracrine mediators that includes various cytokines, chemokines, growth factors and enzymes. These paracrine mediators released into the extracellular space are denoted as the secretome, and these mediators have been associated with the majority of immunomodulatory effects exhibited by MSCs. The secretome has been characterized in multiple studies with the help of proteomics profiling, liquid chromatography-mass spectrometry (LC-MS) and enzyme-linked immunosorbent assay (ELISA). The composition and concentration of the components in the secretome can vary drastically depending upon the cellular source and preparation parameters. The components in the secretome can be categorized majorly as: 1. Soluble mediators and 2. EVs.

### 8.1. Soluble Mediators

#### 8.1.1. IL1 Receptor Antagonist (IL1RA)

A variety of cells, including monocytes, macrophages, and synovial cells, secrete IL-1. Multiple chemokines, cytokines, and inflammatory mediators are induced by IL-1. Numerous studies have documented the role of IL1RA in MSCs’ anti-inflammatory effect. By expressing IL-1RA, MSCs have shown to suppress inflammation in IL1RA knock out mice [97]. MSC-secreted IL1RA has been shown to act on both macrophages and B lymphocytes, preventing the B lymphocytes from differentiating into plasmablasts and encouraging the macrophages to adopt an anti-inflammatory M2 phenotype [98]. Similarly, Phinney et al., showed that MSC secreted IL1RA can protect lung tissue from bleomycin induced inflammation and fibrosis in mice by inhibiting TNF-α and IL-1α, two vital proinflammatory cytokines in the lung produced by activated macrophages [99].

#### 8.1.2. Indoleamine 2,3-Dioxygenase (IDO)

The tryptophan catabolism enzyme IDO is an inducible catalytic rate-limiting enzyme. IDO degrades tryptophan and creates tryptophan-degrading kynurenines, which have an immuno-regulatory impact [100]. IDO is often produced by MSCs in response to inflammatory cytokines, and it has an immunosuppressive impact by polarizing macrophages to the anti-inflammatory M2 phenotype that secretes IL-10 [101]. Song-Guo et al. demonstrated that human gingiva-derived MSCs (GMSCs) reduced inflammatory macrophage activation partly through the IDO/CD73 signal pathways, leading to the conversion of inflammatory macrophages to anti-inflammatory macrophages in an atherosclerosis mice model [102].

#### 8.1.3. Prostaglandin E2 (PGE2)

PGE2 is a complex lipid molecule that is influenced by the cell type and microenvironment in which it is produced. MSC-produced PGE2 helps reduce inflammation and promotes macrophage polarization from M1 to M2 [103]. According to Vasandan et al., *Salmonella enterica* infected macrophages cocultured with MSCs activated respiratory burst and NO-dependent killing mechanisms, boosting macrophage microbicidal activity. MSCs on the other hand, released increased quantities of PGE2 after being treated with IFN-γ, converting M1 macrophages to M2 macrophages. With the help of COX-2 knocked down MSCs (COX-2KD MSCs), the role of PGE2 in inducing M1 macrophages towards an anti-inflammatory M2 phenotype was confirmed. The inducible enzyme COX-2 is involved in the synthesis of PGE2, which when knocked down prevents PGE2 synthesis and inhibits M1 to M2 macrophage polarization [104]. Similarly, in a diabetic cardiomyopathy mouse model, MSC infusion in the presence of high glucose and LPS resulted in increased PGE2 release, which reduced cardiac inflammation by polarizing macrophages from M1 to M2 and secreting IL-10 [105]. The anti-inflammatory properties of PGE2 releasing MSCs were demonstrated once more in an IBD mice model. The study used a chitosan-based injectable hydrogel with immobilized C domain peptide of insulin-like growth factor-1 on chitosan (CS-IGF-1C) and hP-MSCs to reduced inflammation and participated in M1-M2 bioenergetic shifts.

#### 8.1.4. Tumor Necrosis Factor-Stimulated Gene-6 (TSG-6)

TSG-6 is a protein produced during inflammation that has been related to a number of protective and anti-inflammatory properties, including mediating many of MSCs’ immunomodulatory and therapeutic effects [106]. The MSC-derived TSG-6 that is upregulated when MSCs are in direct contact with M1 macrophages reduces T-cell proliferation and pro-inflammatory responses, and may promote the switch from M1 to M2 phenotypes in LPS-induced or spontaneous abortions in mice [88]. In an inflammatory setting, TSG-6 secretion increases. Hongyu Son et al. demonstrated this phenomenon in rats with severe acute pancreatitis, where chorionic plate-derived MSCs produced significant quantities of TSG-6 in a hyperinflammatory environment, repairing pancreatic injury, reducing inflammation and polarizing macrophages from M1 to M2. Similarly, human umbilical cord (UC)-MSCs secreted TSG-6 decreased severe burns and the associated inflammation by inhibiting P38 and JNK signaling [107].

#### 8.1.5. Transforming Growth Factor Beta (TGF-β)

TGF-β is a cytokine that plays a function in immunoregulation and tissue repair [108]. TGF-β works as an immunosuppressive cytokine in MSCs, regulating activated T-cells and macrophages as well as inhibiting the production of iNOS along the SMAD3 pathway in a dose-dependent manner [109]. TGF-β produced by MSCs amid excessive inflammatory responses can cause LPS-stimulated macrophages to polarize to the M2 phenotype, reducing inflammation through the Akt/FoxO1 pathway [110].

#### 8.1.6. Pentraxin 3 (PTX3)

PTX3 is a characteristic acute-phase protein that plays a crucial role in inhibition of inflammation and apoptosis in cells. It is well established that important activators of the inflammatory and reparative response following tissue damage elicit enhanced PTX3 secretion from many different cell types, including MSCs [111]. By treating LPS-stimulated macrophages with PTX3 released from umbilical cord blood (UCB)-MSCs, Kim et al. showed that PTX3 promoted macrophage polarization, leading to decreased inflammation and better anti-inflammatory effects [112]. To a comparable extent, PTX3 secreted by Adipose derived MSCs (ADSCs) also favored an M2 macrophage phenotype and stimulated IL-10 expression in PBMCs isolated from individuals before the onset of metabolic syndrome [113].

#### 8.1.7. Chemokines

MSCs secrete immunomodulatory chemoattractants including C–X–C motif chemokine 12 (CXCL12) [114] and CCL2 [115,116], which have a role in changing the macrophage phenotype to an anti-inflammatory M2 macrophage while down regulating M1-specific markers. Jacques Galipeau et al. demonstrated a synergistic effect of both CXCL12 and CCL2 production by MSCs into its secretome exhibiting an anti-inflammatory environment where tissue resident macrophages are polarized to the M2 phenotype, reducing severe colitis [116].

#### 8.1.8. Mitochondrial Transfer

Mitochondrial transfer has been viewed as a viable therapeutic method since the ability to transport organelles and selective membrane vesicles through extremely sensitive nanotubular structures [117] and also replace damaged mitochondria with healthy mitochondria from an external source [118]. Mitochondria plays an important role in the metabolic reprogramming of the macrophages during their activation [119]. Mitochondrial transfers to macrophages regulate immunomodulatory effects. Maroun Khoury et al. confirmed the immunomodulatory properties of MSC mitochondria in their study of transferring mitochondria to T-cell populations via artificial and natural methods, promoting differentiation into T-regulatory-cells [120]. This approach shows promise to control inflammatory diseases in a mouse graft versus host disease model. Donation of mitochondria by mesenchymal stromal cells causes macrophages to adopt an anti-inflammatory phenotype. Yanrong Lu et al. investigated the mechanisms underlying MSC’s improved mitochondrial function in macrophages (M2) [121] and found that transferring mitochondria from MSCs to macrophages reduced inflammation and reduced kidney injury in a mouse model of diabetic nephropathy by promoting mitochondrial biogenesis via regulating the transcription of PPARGC1A or PGC-1α and clearing out damaged cells via PGC-1/TFEB mediated autophagy [122].

### 8.2. Extracellular Vesicles

It has been shown that MSCs have a paracrine function that goes beyond the release of soluble mediators through the release of EVs. EVs had previously been regarded as inert cellular debris, that was generated as a consequence of cell damage or as a result of dynamic plasma membrane turnover. However, the discovery of EVs’ distinct roles as facilitators of cellular interactions, in which EVs may transport functional molecules to recipient cells and modify their biological and pathological activities, represented a significant milestone in the development of this field of study [123]. EVs are made up of a lipid bilayer containing proteins/peptides, lipids, and genetic material including messenger RNA (mRNA), microRNA (miRNA), and DNA. EVs may also contain ribosomes, proteasomes, and mitochondria [124]. Due to overlapping properties with other nano-sized lipids, protein molecules, and nucleic acid complexes, adequate EV purification and characterization procedures are crucial for drawing exact findings. It is difficult to study EVs because, similar to other nanostructures, they are at or below the detection limit of many standard analytical techniques. The minimal information for extracellular vesicle studies (MISEV) has been crucial in setting the framework and completing the purpose of the International Society for Extracellular Vesicles (ISEV), which is to advance EV science internationally. Recommendations were given at MISEV2018 in six categories, including EV terminology, specimen collection and preliminary processing, EV isolation and concentration, EV characterization, functional research, and reporting necessities/deviations [125].

Following the established standards, researchers are exploring the possibility of MSC-EV-based treatments. EVs derived from MSCs possess many of the same therapeutic effects as MSCs. In a model of silicosis, Phinney et al. discovered that the transfer of mitochondria and miRNAs to human macrophages via EVs improved macrophage bioenergetics while inhibiting Toll-like receptor signaling [126]. Morrison et al. demonstrated that the mitochondria-laden EVs can also alter the phenotype of macrophages in acute respiratory distress syndrome to increase oxidative phosphorylation, phagocytic capacity, and CD206 expression, while decreasing proinflammatory cytokine production in macrophages [127]. Subsequently, EVs generated from MSCs are divided into three primary kinds in order to better understand the biogenesis method, size, and surface markers 1. Exosomes are the smallest vesicles (30–100 nm) released when multivesicular formations, such as intraluminal vesicles (ILVs), fuse with the plasma membrane. 2. Microvesicles are vesicular (0.1–1.0 µm) entities that are shed via plasma membrane blebbing. 3. The largest EVs (1–5 µm) are apoptotic bodies, which are formed at the late stages of apoptosis [128]. Due to their nanoscale dimensions and excellent protective effects, exosomes have received the most research interest in recent years.

#### 8.2.1. Exosomes

Exosomes are nanosized vesicles of endocytic origin. Their development is influenced by the process of endocytosis and exocytosis. Exosome development begins when an early endosome absorbs a small amount of intracellular fluid [129]. When an early endosome evolves into a late endosome, an intraluminal body, also known as a multivesicular body (MVB), is produced. Exosome production is dependent on multiple biological pathways [130]. Endosomal sorting complex required for transport (ESCRT) is one of them; it possesses both ESCRT-dependent and ESCRT-independent transport pathways. The ESCRT apparatus includes ESCRT-(0, I, II and III) complexes [131]. As MVBs grow from the early endosome, ESCRT-dependent/-independent machinery forms ILVs. Following transport and fusion with the plasma membrane by MVBs, the ILV containing exosomes are released [129]. Exosomes have been demonstrated to perform numerous functions, including pro-angiogenic, pro-tumorigenic, anti-clotting, anti-inflammatory and immune modulation [132]. MSC exosomes induce immunosuppressive effects by encouraging the formation of anti-inflammatory M2 macrophages and inhibiting the synthesis of pro-inflammatory proteins [133,134]. Exosomes produced from MSCs have been proven in multiple studies to prevent macrophage polarization and recruitment. Stimulated by MSC exosomes, colonic macrophages adopt an immunosuppressive M2 phenotype [135]. By promoting the formation of M2 macrophages, MSC exosomes can reduce inflammation, and ultimately promote wound healing and tendon repair [136]. MSC exosomes can also alleviate alveolar damage and inflammation in the lungs by boosting the M2 macrophages as a proportion of total macrophages [72] ADSCs exosomes produce M2 macrophages through the S1P/SK1/S1PR1 signaling pathway which is protective against cardiac apoptosis and fibrosis [137]. The primary contribution of UC-MSCs derived exosomes to spinal cord injury recovery was inducing macrophages to change from an M1 to an M2 phenotype, where the M2 markers were upregulated and the M1 markers such as TNF-α and iNOS levels decreased [135]. Increased CD163^+^ regenerative M2 macrophages and decreased CD86^+^ M1 macrophages were observed in the osteochondral defect and surrounding synovium following treatment with exosomes, and levels of synovial pro-inflammatory cytokines, including IL-1β and TNF-α, were lowered [138]. MSC exosomes transformed monocyte-derived myeloid-derived suppressor cells into M2 macrophages by guiding the expression of programmed death-ligand 1 (PD-L1) in macrophages with the help of TGF-β and semaphorin [139]. MSC exosomes can also increase matrix production and deposition during cartilage regeneration [140,141,142,143]. According to these studies, the matrix-degrading enzymes such as MMP-13 [144,145,146] or ADAMTS-5 [147] were simultaneously inhibited in vivo and increased cartilage regeneration. All of these findings suggest that MSC exosomes may influence the tissue immune cells to adopt a regenerative immune phenotype that is favorable for tissue repair and regeneration. 

#### 8.2.2. Exosomal miRNA

MSC exosomes contribute to the immunosuppressive effects on macrophages by way of miRNA. The miRNAs utilized by MSC exosomes to promote the anti-inflammatory phenotype of macrophages in various inflammatory models are listed in Table 2. Ragini et al. conducted a detailed characterization of human amniotic membrane-derived MSCs (hAMSCs)-secreted EVs, and they identified the presence of 200 secreted factors and 754 miRNAs. Of the miRNAs, five miRNAs are involved in M2 (miR-24-3p, miR-146a-5p, miR-222-3p, and miR-34a-5p), and three involved in M1 (miR-125b-5p, miR-145-5p and miR-130a-5p) phenotype regulation [148]. There is mounting evidence that aging affects the physiology of both macrophages and MSCs. Huang et al. demonstrated that young MSC exosomes exhibit high levels of miR-223-5p expression, whereas older MSC exosomes exhibit miR-127-3p and miR-125b-5p expression. Younger MSC exosomes are better able to convert M1 macrophages to M2 than older MSC exosomes [149]. The anti-inflammatory miR-146a derived from MSC exosomes facilitates the transition from M1 to M2-like macrophages [150]. Co-cultivating MSC exosomes with M1 macrophages significantly reduced the production of pro-inflammatory M1-like markers, including IL-6, TNF-α, and CCL5 [151]. MSC exosomes can reduce neural inflammation by upregulation CD206 and Arg1 to regulate the polarization of glial cells and macrophages. In addition, they can block the synthesis of pro-apoptotic proteins, such as Bcl-2 related X protein (BAX) and iNOS, as well as pro-inflammatory cytokines [152]. Overexpression of miRNAs such as miR-140-5p and miR-92a-3p has been associated with anti-OA effects via increased matrix formation and decreased cartilage degradation [145,153]. Other studies have demonstrated that suppression of miR-100-5p or lncRNA KLF3-AS1 can increase the protective effects of MSC exosomes against matrix breakdown during OA [144,146]. In conclusion, MSC exosomes modify the polarity of macrophages and affect factors associated with this process, allowing for improved immunosuppressive regulation and cartilage regeneration. However, macrophage polarization in OA models employing MSC-derived paracrine mediators is poorly understood. The precise mechanism of their impacts on OA treatment, including the role of each miRNA and protein, is not entirely understood. Therefore, more in-depth research is necessary to understand the essential internal components of the secretome and their molecular processes on OA. In this review, we explored the involvement of numerous paracrine mediators in various inflammatory disease models in order to give knowledge that may be used to build a validated mechanism for OA that is comparable to the MSC biology and macrophage interaction reported in other inflammatory models.

## 9. Priming Enhances Mesenchymal Stem Cell Immunomodulation

The significance of paracrine signaling in MSC therapy, particularly with relation to the immunomodulatory properties of MSCs, is becoming increasingly apparent. This emphasizes the need to modify the secretory profiles of MSCs during in vitro production in order to achieve desired functional properties. This can be accomplished by regulating the secretion of individual immune mediators or by enhancing the secretome’s overall output. Priming MSCs with pro-inflammatory cytokines, pharmacological agents, or small molecules, or by the application of biophysical priming techniques, has increased the release of specific anti-inflammatory and immune regulatory factors [167]. 

### 9.1. Proinflammatory Cytokines

Priming MSCs with proinflammatory cytokines such as TNF-α has been demonstrated to upregulate key paracrine mediators such as IDO, PGE2, and hepatocyte growth factor (HGF), but to a lesser extent than IFN-γ. MSCs primed with IFN-γ have been reported to release adhesion proteins VCAM1 and ICAM1, as well as the chemokine ligands CXCL9, CXCL10, and CXCL11 and high levels of HLA-G and IDO. Co-culturing IFN-γ primed MSCs with activated PBMCs increased the frequency of CD4^+^CD25^+^CD127^dim/-^ T-cells, IL-10 and IL-6, while decreasing the frequency of Th17 cells, IFN-γ and TNF-α production [168]. MSCs stimulated with IFN-γ enhanced production of programmed cell death-1 ligands (PDL-1) to inhibit T-cell effector function and have also been demonstrated to inhibit NK cell cytotoxicity [169]. Similarly, BM-MSCs preconditioned with IL-17 reduced Th1 secretion of cytokines such as TNF-α, IFN-γ, IL-2 and enhanced iTreg cell formation. Furthermore, genes such as MMP1, MMP13, and CXCL6 that are predominantly correlated with migration and chemostatic responses were identified [170].

In light of the fact that MSCs from varying donors and sources exhibit varying cytokine priming responses [171], it may be necessary to combine cytokine priming in order to maintain a significant and consistent effect. Compared to priming MSCs with a single agent, combined IFN-γ and TNF-α priming significantly reduced donor-specific variability in MSC immunomodulatory potency. Chenyang Liu et al. demonstrated that supernatant from MSCs that have been pretreated with IFN-γ and TNF-α has been shown to switch macrophages to the M2-type, which in turn promotes cutaneous wound recovery with minimal scarring by stimulating the IL-6-dependent signaling pathway [172]. Likewise, in another study MSCs isolated from menstrual blood and stimulated with IFN-γ and TNF-α showed elevated levels of IDO1, EV release, and differential expression of miRNAs related to the immune response and inflammation [173]. Besides IFN-γ and TNF-α, MSCs primed with different combinatorial cytokine cocktail like LPS/ TNF-α also exhibited polarization of macrophages to the M2 phenotype expressing high amounts of PGE2, Arg1, and CD206, and displayed improved alkaline phosphate activity and bone mineralization potential [174]. The miRNA expression profile of foreskin MSCs is drastically altered after treatment with a cytokine cocktail containing IL-1β, TNF-α, IFN-γ, and IFN-α, with 13 miRNAs being downregulated and 3 others being upregulated. These miRNAs with altered expression levels are speculated to target multiple potential signaling pathways that control cellular activity in response to inflammatory cues. Several pro-inflammatory cytokine mixes have been used to alter the expression of immune mediators and miRNAs by MSCs in culture [175]. One of the major drawbacks of this strategy is the cost of recombinant cytokines.

### 9.2. Chemical Agents

In order to reduce the costs associated with recombinant cytokines, MSCs have been primed with a variety of pharmacological chemicals and small molecules in order to increase their therapeutic efficacy. The use of chemical agents like all-trans retinoic acid (ATRA) has shown to inhibit PBMC production of pro-inflammatory cytokines. Priming MSCs with ATRA improves wound-healing capacities in vivo. The gene expression of COX-2, VEGF, CCR2, HIF-1α, CXCR4, angiopoietin-2 (Ang-2), and angiopoietin-4 (Ang-4) is elevated by preconditioning rat BM-MSCs with ATRA [176]. Matteo Haupt et al. showed that the therapeutic potential of MSC EVs preconditioned with lithium is higher than that of EVs from native MSCs. Increased levels of miR-1906, a new regulator of toll-like receptor 4 (TLR4) signaling, were found in MSC EVs after treatment with lithium, which led to reduced cerebral inflammation and rapid neuroprotection in mice with stroke [177]. The histone deacetylase inhibitor valproic acid (VPA) and the bioactive lipid sphingosine-1-phosphate (S1P) have similar anti-inflammatory and proliferative actions. Priming MSCs with valproic acid and lithium before intranasal infusion improved neuropathological characteristics and function in a mouse model of Huntington’s disease [178]. Priming MSCs with cytokines and chemicals added to MSC culture media facilitates their ex vivo growth for therapeutic applications. This has been examined in relation to the preparation of xeno-free and serum-free media. In a recent study, Jin et al. created a hypoxic, calcium-rich environment for stem cells to grow while preserving them in a xeno-free, chemically defined cryopreservation media. The paracrine factor PTX-3 generated by these stem cells was shown to remodel M1 macrophages into their anti-inflammatory M2 phenotype in a rat OA model [179]. Small molecules are also being investigated as a method of priming MSCs due to their unique qualities such as low cost, minuscule size, robust stability, and non-immunogenicity. Oren Levy et al. showed a decrease in the expression of TNF-α at the site of inflammation after pre-treatment of MSCs with a kinase inhibitor (Ro-31–8425). Similar to Ro-31-8425, priming of MSCs with the small molecule tetrandrine boosted PGE2 synthesis via the NF-κB/COX-2 signaling pathway, which reduced TNF-α production in RAW264.7 during co-culture [180].

### 9.3. Hypoxia

Under hypoxic growth conditions, when the oxygen level is between 0 and 10%, MSCs can secrete more immunomodulatory molecules. It is well documented that hypoxic preconditioning can stimulate the production of immunomodulatory molecules in MSCs, such as IDO, IL-10 and PGE2 [181]. Hypoxia-exposed MSCs drive bone marrow-derived macrophage polarization to the M2 phenotype via the TGF-1/Smad3 signaling pathway, ameliorating ischemic stroke conditions by reducing apoptotic cells and fibrosis and promoting neovascularization in the infarcted region [182]. According to recent studies, EV density and load might also be modified by employing hypoxic preconditioning. However, hypoxia seemed to have no effect on the mean size, morphology, or surface biomarkers of MSC-derived EVs [183]. In response to hypoxia and serum deprivation, primed MSCs produced more dipeptides, suggesting that hypoxic MSCs augment their pool of free amino acids to meet energy requirements that cannot be properly met by the glycolytic process. Subsequently, it was also established that there are 21 different metabolites in primed MSC derived exosomes that have been linked to immunoregulation. The activation of regulatory T-cells, the polarization of macrophages toward the M2 state, and the regulation of anti-inflammatory responses are all directly influenced by these molecules [184]. Despite evidence indicating that MSCs grown under hypoxic conditions can result in the production of EVs, the real situation is still unclear. The variation may be attributed to the degree of hypoxia, as minute variations in oxygen concentration and exposure time can have a significant impact. In addition, it is important to note that while some studies have shown that hypoxia may promote cellular longevity, others have shown that cells may die [185].

### 9.4. Biophysical Stimulation

Another strategy that has been investigated is biophysical stimulation of MSCs. Priming approaches, including altering the texture and rigidity of culture surfaces, may influence cytokine release by MSCs [186]; however, this method has limited scalability. In an effort to create an environment that is analogous to that of the MSC niche, researchers have investigated the use of a variety of 3D based cell culture approaches [187]. When maintained in a three-dimensional environment, MSCs tend to produce more immunomodulatory factors. Spheroid creation is the most popular approach for MSC 3D cultivation [188]. Under these conditions, less oxygen may diffuse into the inner layer of cells, creating a hypoxic environment that enhances cell-cell interactions and modifies the release of immunomodulatory molecules. MSCs secreted more TSG6, HGF, and PGE2 when cultured in 3D spheroids; IDO activity and the ability to limit T-cell proliferation were both found to be attenuated when MSCs were cultured in aggregates [189,190]. Hydrogel encapsulation of MSCs is one of the most exciting approaches for producing a 3D-MSC-secretome. Hydrogels permit the change of the mechanical properties such as rigidity and firmness and the inclusion of patterns unique to the natural ECM, both of which increase the secretome’s complexity. Recent attention has been drawn to biopolymer hydrogels due to their capacity to alter the paracrine actions of MSCs [191,192]. The field of cell engineering is expanding fast, making all these methods particularly attractive.

## 10. Effect of Macrophages on Mesenchymal Stem Cells

We know that the phenotype of macrophages can alter upon interaction with MSCs. Similarly, macrophages also have a feedback impact on MSCs that affects their migration, viability, differentiation, and immunomodulatory capabilities. Guihard et al. demonstrated that conditioned media from human monocytes activated with LPS or TLR ligands promoted bone formation by human BM-MSCs [193]. M1 macrophages promote osteogenesis in MSCs via stimulating the COX-2-PGE pathway [194,195,196]. Regardless of their polarization status (M0, M1, or M2), human ADSCs can be blocked from transforming into adipocytes in vitro by macrophage derived supernatants [197]. According to previous studies, M2-type macrophages enhance MSC proliferation and migration, but M1-type macrophages cause MSC apoptosis [198,199]. According to de Witte et al., the phagocytosis of MSCs by monocytes is essential for the immunological regulation of MSCs [200]. Li et al. discovered that enhanced synthesis of TSG-6 in response to contact with pro-inflammatory macrophages improves MSCs’ inhibitory control of T-cells and macrophages [88]. Mouse BM-MSCs cocultured with macrophages enhanced IL-10 release in response to LPS stimulation via a PGE2-dependent mechanism. MSCs cannot secrete PGE2 under coculture conditions unless activated by TNF-α and iNOS generated by macrophages [201]. In response to pro-inflammatory cytokines produced by macrophages, MSCs produce immune modulators such as PGE2 and IL-1RA [202]. According to the aforementioned research, macrophages produce cytokines that activate MSCs after being activated by pro-inflammatory mediators MSCs respond to the activation of macrophages by modulating the immune response. There is a feedback loop between macrophages and MSCs within the disease microenvironment. MSCs and macrophages work together to keep the inflammatory environment in balance.

## 11. Challenges and Future Perspectives

Clearly, macrophages are crucial for both homeostasis and disease pathology. In this review, we have focused on the role of macrophages in OA inflammation with a particular emphasis on the role of macrophages in the synovium, as the synovium is the predominant site of gross and microscopic inflammatory change in OA. Synovial macrophages are increasingly believed to contribute to the development and persistence of inflammation in OA, based on data from research including human patients and animal models. The reprogramming of macrophages has therefore been recommended in this review for prevention against synovitis and cartilage destruction in OA. However, a number of unsolved concerns complicate the development of macrophage-targeting therapies for OA. First, macrophage transcriptome research has demonstrated the need for a more in-depth evaluation of macrophage functional characteristics, showing the limitations of the M1/M2 paradigm. Second, the identification and verification of biomarkers that are specific to different macrophage subgroups is essential to the comprehension of macrophage diversity in OA as well as the development of therapeutic alternatives that are connected to the specific immunological profile. Lastly, the engagement of macrophages in the pathogenesis of OA may differ depending on the stage of the condition and the endotype, and the inflammatory mediators and mechanisms vary greatly across individuals. Encouragingly, recent advances in technology, including as single-cell RNA sequencing and mass cytometry, have made it possible to precisely identify the cellular diversity of macrophages. In actuality, a two-phase strategy comprising the characterization of the functional implications of macrophage sub-populations and the development of particular targeted techniques applicable for macrophage remodeling may be essential.

With respect to the second phase of the two-phase strategy, we have highlighted recent efforts to target macrophage activity in OA using small molecules and biologics which have the ability to modulate the inflammation. Despite the fact that all bioactive substances and chemical compounds have been demonstrated to target OA-related signaling pathways and effectively polarize macrophages, further research is necessary. There are various obstacles to overcome before herbal bioactive components may be used in clinical settings. Due to the limited stability and low absorption of bioactive compounds in serum or synovial fluid, the therapy for OA is accompanied by a number of serious side effects. Second, regulated extraction methods cannot compensate for the fact that the quality of raw herbs varies, and it is difficult to standardize the concentration of the principal ingredients. Lastly, to confirm the safety of the compounds for clinical use, rigorous and standardized toxicity studies will be required. In terms of nanoparticle-based approaches, there are still a number of obstacles that must be surmounted before it may be used in human clinical practice. In order to reduce discomfort and improve patient compliance, it is crucial that the nanoparticles dissolve at a set time. Furthermore, the optimal dose, quantity, frequency, and timing of treatment must be established based on the severity and location of the condition. Finally, the inability of bioactive chemicals and nanoparticles to regenerate the cartilage is a fundamental limitation of their usage.

Recently, there has been a lot of interest in the use of MSCs and its secretome for treating OA, in the realms of regenerative medicine and tissue engineering. In light of this, we have emphasized the significance of MSC-derived secretome in polarizing macrophages in diverse models of inflammation, as well as the strategies used to enhance the immunomodulatory potential of MSCs and their paracrine mediators, which give novel insights that may be used for OA research. These insights can be used to develop a gold standard mechanism for OA that closely resembles the MSC biology and macrophage interaction observed in other inflammatory models. MSC’s capacity for natural cartilage repair and regeneration, as well as its ability to modulate macrophage phenotype, is an advantage of employing MSCs to treat OA above all other polarization approaches. Several studies have demonstrated that injection of MSCs stimulates cartilage tissue regeneration, due to the polarization of macrophages [203,204]. There are still certain restrictions to employing MSCs for therapeutic reasons, despite extensive study and remarkable advancements. One of these obstacles is the absence of standardized protocols. Variations in cell source, cell isolation and culture processes, or administration routes are frequently cited as the cause of disparities in reported results [205]. Concerningly low rates of cell retention and survival following implantation are a further factor to consider. It has been demonstrated that less than 1% of MSCs survive more than a week following systemic administration [206]. This therapy window may not be sufficient for the overwhelming majority of individuals [206,207]. Therefore, attention has switched from these difficulties to the MSC-derived secretome since the secretory activities of MSCs are believed to be the underlying rationale for their therapeutic properties [206,207].

Several studies demonstrate that the immunomodulatory actions of MSCs are mostly attributable to their secretome, which has led to a turning point when cell-based therapies could be substituted. In this regard, the immunomodulatory/anti-inflammatory secretome produced from MSCs is the new gold mine, bypassing the limitations of cell-based therapies. This motivates the scientific community to investigate its medicinal properties, and interest in the topic is fast growing. It has been established that employing the secretome as a treatment for OA produces therapeutic outcomes [164,208,209]. However, there is still a long way to go in terms of study, as this therapeutic potential has yet to transcend its constraints. First, it is important to characterize the secretome in depth so that its applications may be studied and reproduced. There are indications of a changing composition of the secretome depending on both external and internal conditioning variables of the source cells. The secretome’s positive effects might also be amplified by priming of MSCs. In this review, we have focused on the methods currently in use to prime MSCs in an effort to boost the secretomes’ immunomodulatory capacity. Priming techniques of MSCs have significant difficulties in clinical translation, including induction of immunogenicity, increased costs, unpredictable results, and a lack of clinically applicable good manufacturing procedures (GMP). In order to do the spadework for different priming approaches in the clinical setting, we need to evaluate (1) Optimal sources for isolating MSCs, (2) Epigenetic modifications, (3) Antigenicity and tumorigenicity of primed and non-primed MSCs, and (4) Appropriate good manufacturing practices (GMP) standards for quality control of MSC products [210]. It is also generally accepted that the functional properties of primary MSCs might vary depending on the age, anatomic origin, and in vitro expansion of donors. Consequently, their secretome profiles may vary substantially [211]. By utilizing standardized MSCs derived from induced pluripotent stem cells, it is possible to circumvent these limitations and expand their therapeutic utility. Similarly, there is no consensus on the optimal approaches to standardize and personalize the secretome content, which is required for the development of medicines with a diverse range of applications. The development of finely tuned procedures for secretome extraction is a further objective that has not yet been reached. According to the available literature, we may propose two methods for modifying for enhancing MSC-derived secretome performance. (1) Standardization of approaches for isolating and purifying the entire secretome with maximum yield and scalability; and (2) Development of secretome delivery protocols and dosages.

## 12. Conclusions

Current OA treatment outcomes place a significant strain on worldwide healthcare systems. Deterioration of the articular cartilage results in an aberrant immune microenvironment in the joint. This dysfunction of the local microenvironment leads to a wide variety of uncontrolled inflammatory responses. Once inflammatory processes have occurred, OA can be treated by restoring balance to the local immunological environment in the joints. Controlling OA requires polarizing macrophages, a natural component of the joint’s local immunological milieu. Although macrophages are a viable therapeutic target, it is necessary to first comprehend their phenotypes in terms of their characteristics, anatomical location, and origins. The microenvironment influences the phenotype and function of macrophages. Restoring a healthy equilibrium necessitates an in-depth examination of the anatomical setting in which macrophages operate. Modifying the phenotype of macrophages to influence the development, progression, and resolution of inflammation by acting on molecules in signaling pathways and the local microenvironment is a potential field for the treatment of OA.

Despite the fact that MSCs can induce macrophage polarization toward the M2 phenotype, there are a number of clinical risks and complications associated with this cell-based therapy. The MSC-derived secretome, which carry the majority of MSCs’ therapeutic effects, represent a novel therapeutic strategy. This cell-free therapy circumvents the disadvantages of MSCs and has certain advantages. But there is a need for the creation of guidelines to improve experimental conditions for producing MSCs’ secretome, to establish more standardized protocols among the scientific community, and to encourage future collaborative work to bridge the decades-long gap between MSCs’ experimental research and their clinical use.

## Figures and Tables

**Figure 1 ijms-23-13016-f001:**
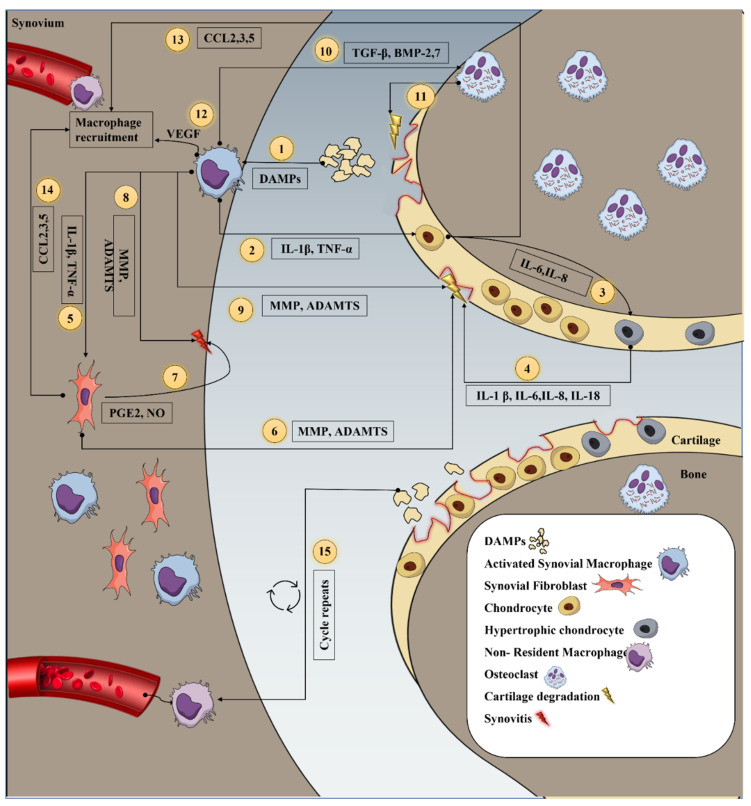
Role of macrophages in OA inflammation. Macrophages respond to DAMPs, such as necrotic cell proteins and cartilage fragments, by releasing a variety of pro-inflammatory mediators (1). Macrophage IL-1β and TNF-α encourage chondrocytes to secrete more IL-6 and IL-8 (2,3), leading to a hypertrophic state and the release of more pro-inflammatory mediators that damage the cartilage (4). IL-1β and TNF-α also stimulate synoviocytes to release pro-inflammatory mediators (5), such as MMPs, ADAMTS, PGE2, and NO, that cause cartilage degradation (6) and synovitis (7). Macrophages also release various MMPs and ADAMTS that cause synovitis (8) and cartilage breakdown (9). Macrophages release TGF-β and BMP-2,7 that promote osteoclast development (10), which further degrades the cartilage and bone (11). VEGF produced by macrophages (12) and CCL2,3,5 produced by chondrocytes (13) and synoviocytes (14) help recruit more macrophages to the joint, repeating steps 1–14 (15).

**Figure 2 ijms-23-13016-f002:**
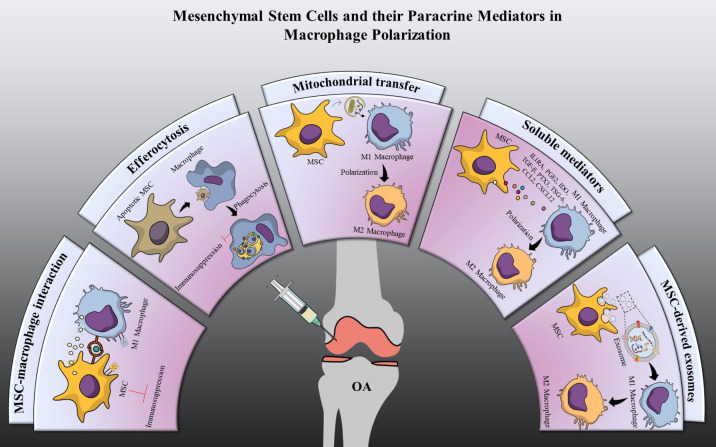
Schematic representation of mesenchymal stem cells (MSCs) and their paracrine mediators in macrophage polarization. MSCs exert immunosuppressive effects by switching pro-inflammatory M1 macrophages to anti-inflammatory M2 macrophages through (1) MSC-macrophage interaction: Immunosuppression is triggered by adhesion receptors and calcium signaling following a functional contact between MSC and an inflammatory M1 macrophage. (2) Efferocytosis: Apoptosis and efferocytosis of MSCs induces metabolic and inflammatory alterations in macrophages, resulting in immunosuppression. (3) Mitochondrial transfer: Induces functional alterations in macrophages and improves the immune-regulatory activity by M2 macrophage activation. (4) Soluble mediators: Enhances macrophage immunomodulation by reducing pro-inflammatory M1 state and maintaining anti-inflammatory M2 phenotype. (5) Exosomes: The exosomal cargo and molecular contents have the capacity to influence the macrophage phenotypes.

**Table 1 ijms-23-13016-t001:** Therapeutic strategies for macrophage polarization.

No.	Compound/Target	Experimental Model	Mode of Administration and Dosage	Major Findings	Signaling Pathway	Reference
1.	SHP099	DMM mouse model	Intra-articular 10 μL of 20 μmol/L SHP099	Allosteric src-homology 2-containing protein tyrosine phosphatase 2 (SHP2) inhibitor SHP099, decreased joint synovitis, reduced M1 macrophage polarization, increased COL2, and decreased COL10 and MMP3 in SHP2 knockout mice.	NF-κB and PI3K pathway	[58]
2.	The meta defensome	CIOA mouse model	Intravenous 100 μL of 1 mg/mL meta defensomes	Reprogrammed the mitochondrial (mt) metabolism of M1 macrophages by scavenging mtROS, inhibiting mtNOS and polarized M1 to M2 phenotype via regulating the expression of mitochondrial transcription factor A (TFAM).	Reprogramming metabolic pathway of M1 macrophages	[59]
3.	Fargesin	CIOA mouse model	Intra-articular 5, 10 or 20 mg/kg body weight (BW)	Increased macrophage polarization and decreased apoptotic chondrocyte and activated macrophage crosstalk in the early stages of OA.	p38/MAPK/NF-κB signaling pathways	[60]
4.	Angelicin	DMM mouse model	Intraperitoneal 20 mg/kg BW	Polarizes M1 macrophages to M2 phenotype in the synovial tissues and protective of maintaining the M2 phenotype. Protects the cartilage from damage.	CD9/gp130/STAT3 pathway	[61]
5.	α-defensin-1	Meniscal/ligamentous injury (MLI) rat model	Intra-articular 250 µL (10 ng/mL)	α-defensin-1 reprograms macrophages from M1 to M2 phenotype, the polarized M2 macrophage mediates the reprogramming of pro-catabolic chondrocyte to anabolic chondrocyte.	Insulin signaling and Toll-like receptor (TLR) pathway	[62]
6.	Frugoside	CIOA mouse model	Intra-articular 0.2 mg/kg BW	Prevents polarization of synovial macrophages to M1 macrophages by downregulating miR-155 levels. Helps to delay cartilage degradation and reduces chondrocyte hypertrophy and ECM degradation.	Regulation of miR-155	[63]
7.	Pinosylvin	In vitro murine J774 macrophages and human U937 monocytes	In vitro 10 µM, 30 µM, 60 µM	Suppressed M1 related markers (NO, IL-6, MCP-1, p65 and JNK) and polarized the macrophages to produce M2 markers (Arg-1, Ym1, MRC1, PPARγ and STAT6).	NF-κB and JNK pathway	[64]
8.	Transient receptor potential vanilloid 1 (TRPV1)	Radial transection of the medial meniscus rat OA model	Intra-articular 50 μL of 50 μM capsaicin (CPS)	Blocking TRPV1, a potential therapeutic target for macrophage polarization using CPS (agonist of TRPV1) attenuated joint swelling, improved the synovitis score, reduced M1 macrophage levels, decreased cartilage degeneration and osteophyte formation.	Ca^2+^/calmodulin-dependent protein kinase II (CaMKII)/ nuclear factor erythroid 2–related factor 2 (Nrf2) pathway	[65]
9.	Resolvin D1-loaded nano liposome	Destabilization of the medial meniscus (DMM) mouse model	Intra-articular 1 mg/10 µl	Promoted the resolution of inflammation by increasing the proportion of M2 macrophages in the synovium. The controlled release of resolvin D1 alleviated OA symptoms such as osteophyte formation, cartilage damage and OA associated pain.	Acts on formyl peptide receptor 2 (ALX/FPR2)	[66]
10.	Zeolitic imidazolate framework-8 (ZIF-8) nanoparticles (NPs)	ACLT mouse model	Intra-articular 20 µL of 1 mg/ml	ZIF-8 NPs modified with anti-CD16/32 to target M1 macrophages and the encapsulated S-methylisothiourea hemisulfate salt and catalase inhibited NO and H_2_O_2_ production and induced O_2_ production which improved the mitochondrial function. Hypoxia-inducible factors-1α (HIF-1α) was inhibited and prevented chondrocyte hypertrophy in vitro and cartilage degeneration in vivo.	MAPK and NF-κB pathway	[67]
11.	Quercetin	Removal of medial meniscus and the anterior meniscotibial ligament	Intra-articular 8 µM (100 µL/joint cavity)	Induces the M2 phenotype in synovial macrophages, hence reducing inflammation and apoptosis and stimulating chondrocyte glycosaminoglycan synthesis to aid in the repair of destroyed cartilage.	Akt/NF-κB signaling pathway	[68]
12.	Kinsenoside	ACLT mouse model	Intraperitoneal 2.5, 5, 10 mg/kg BW	Plays a multifunctional role by attenuating the infiltration of M1 macrophage, promote polarization of M1 macrophage to M2 phenotype, reduce macrophage conditioned medium and IL-1β induced articular cartilage degeneration and chondrocyte apoptosis.	NF-κB/MAPK pathway	[69]
13.	Marine squid type II collagen (SCII)	ACLT mouse model and meniscectomy (pMMx) rat OA model	Intra-articular 10 mg/mL (100 µL/joint cavity)	Mediated phenotypic shift from M0 to M2 in macrophages. Suppressed apoptosis and hypertrophy in chondrocytes and increased the pro-chondrogenic and ECM related markers.	STAT6 pathway	[70]
14.	R-spondin 2 (Rspo2)	CIOA and DMM mouse model	Intra-articular	Anti-Rspo2 antibody was used to effectively reduce the cartilage degeneration incurred by M1 macrophages that secrete high amounts of Rspo2 and increased the expression of cartilage matrix components (SOX-9, COL2A1, aggrecan).	mTORC1 pathway	[55]
15.	Triamcinolone acetonide (TA)	Rat model of severe OA	Intra-articular 100 µg TA/70 µl	TA enhanced the expression of folate receptor beta (FRβ+) in macrophages and fully prevented osteophyte development in vivo. Also induced differentiation of monocytes towards anti-inflammatory M2 phenotype resulting in the increase in expression of IL-10 in vitro.	Regulates FRβ expression	[71]

**Table 2 ijms-23-13016-t002:** MSC-derived exosomes polarize macrophages to decrease inflammation in various inflammation models.

No.	MSC Source	Exosome Inclusion	Mode of Administration and Dosage	Major Findings	Signaling Pathway	Disease Model	Reference
1.	Human UC-MSCs	miR-146a-5p	Intravenous 2 × 10^6^/500 μL UC-MSCs	miR-146a-5p targeted the TRAF6-STAT1 pathway to suppress kidney inflammation and restore renal function by increasing M2 macrophage polarization.	TRAF6-STAT1 pathway	Streptozotocin-induced diabetic nephropathy rat model	[154]
2.	Human ADSCs	miR-451a	Implanted 0.8 mg exosomes/1 mL PBS	Targeting macrophage migration inhibitory factor, mir-451a can suppress inflammation and induce the polarization of M1 macrophages to M2 macrophages. Exosomes encapsulated in gelatin nanoparticles hydrogel can precisely reach their targets and exert their effects.	Macrophage migration inhibitory (MIF) downregulation	Skull defect rat model	[155]
3.	Mouse BM-MSCs	miR-21a-5p	Intravenous 200 μL (0.5 mg/mL) MSC exosomes	miR-21a-5p inhibits the KLF6 and ERK1/2 pathways, preventing macrophage invasion and promoting macrophage polarization to M2.	MAPK and Akt pathway	Atherosclerosis mouse model	[156]
4.	Mouse MSCs	miR-21-5p	Intramyocardial 50 μg/25 μL MSCs exosomes	miR-21-5p promotes macrophage polarization to the M2 phenotype, which reduces inflammation and facilitates cardiac repair.	5p/TLR4/PI3K/Akt signaling pathway (yet to be confirmed)	Myocardial ischemic injury mouse model	[157]
5.	Human MSCs	tsRNA-21109	In vitro	tRNA-derived fragments (tRFs) polarize macrophages toward the M2 phenotype.	Rap1, Ras, Hippo, Wnt, MAPK, and TGF-β signaling pathways	In vitro	[158]
6.	Human BM-MSCs, Jaw JM-MSCs	miR-223	Intravenous 2 × 10^6^ cells/mL (BMMSC group); 2 × 10^6^ cells/mL (JMMSC group); 200 μg/200 μL BMSCs exosomes	Blocks the pknox1 gene, which is implicated in the activation of M1 proinflammatory macrophages and causes polarization from M1 to M2, resulting in cutaneous wound healing and tissue restoration.	pknox1 downregulation	Skin excised mouse model (cutaneous wound)	[159]
7.	TNF-α preconditioned human GMSCs	miR-1260b	Intravenous 200 μg/200 μL GMSCs exosomes	TNF-α increased M2 macrophage polarization via boosting CD73 expression on exosomes, hence reducing inflammation and halting bone loss in periodontal tissue. miR-1260b was necessary for osteoclastogenesis inhibition.	Wnt5a-mediated RANKL pathway	Ligature-induced periodontitis mouse model	[160]
8.	Mouse BM-MSCs	miR-182	Intramyocardial 50 μg/25 μL BM-MSCs exosomes	miR-182 polarizes macrophages to M2 phenotype within the heart through activating the PI3K/Akt pathway and reduces inflammation by negatively regulating the TLR4 mediated NF-κB pathway.	TLR4/NF-κB and PI3K/Akt signaling pathway	Myocardial ischemia-reperfusion mouse model	[161]
9.	Mouse BM-MSCs	miR-216a-5p	Intravenous 200 μg/200 μL hypoxia induced exosomes	miR-216a-5p extracted from hypoxic MSCs can decrease microglial-induced neuroinflammation by increasing microglial polarization from M1 to M2 through activating the PI3K/Akt pathway and by blocking the TLR4 signaling pathway.	TLR4/NF-κB/PI3K/Akt	Spinal cord injury mouse model	[162]
10.	Mouse ADSCs	miR-let7	Intravenous 100 μg/200 μL ADSCs exosomes	miR-let7 inhibits the high mobility group A protein 2 (HMGA2), which promotes the release of pro-inflammatory cytokines via the NF-κB pathway while simultaneously suppressing macrophage infiltration via the IGF2BP1/PTEN pathway.	miR-let7/HMGA2/NF-κB pathway and miR-let7/IGF2BP1/PTEN pathway	Atherosclerosis mouse model	[163]
11.	TGF-β1 treated rat BM-MSCs	miR-135b	Intra-articular 1 × 10^11^ exosome particles/ml	miR-135b inhibited the degradation of cartilage tissues by increasing the polarization of macrophages to the M2 state and inhibiting MAPK6 expression.	MAPK6 downregulation	OA rat model	[164]
12.	Rat BM-MSC	N/A *	Intra-articular 10^10^ exosomes particles/ml	Exosomes increased the differentiation of synovial macrophages from M1 to M2, reduced chondrocyte hypertrophy and the damage to articular cartilage, delayed the progression of OA, and enhanced joint function.	N/A	OA rat modified Hulth model	[165]
13.	Human UC-MSCs	has-miR-122-5p, has-miR-148a-3p, has-miR-486-5p, has-miR-let-7a-5p, and has-miR-100-5p	Intra-articular 80 μg/ml	Reduced OA progression by transferring important miRNAs to control the PI3K-Akt pathway and polarize M2 macrophages, which affects inflammatory and immunological reactivity.	PI3K-Akt pathway	ACLT OA rat model	[166]

* N/A-Information not available.

## Data Availability

Not applicable.

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
