# Peer review of "Role of Mesenchymal Stem Cells and Their Paracrine Mediators in Macrophage Polarization: An Approach to Reduce Inflammation in Osteoarthritis"

_ijms, 2022, doi:10.3390/ijms232113016_

Round 1

Reviewer 1 Report

The authors have summarized role of mesenchymal stem cells and their paracrine mediators in macrophage polarization. Macrophage polarization is very common in the inflammation of osteoarthritis. MSCs therapy play an important role in the treatment of osteoarthritis due to their multiple differentiation, immune regulation, anti-inflammatory and other characteristics. It’s appealing, and some issues must be addressed:

1.     There are too many titles for the sections, which need further integration. 

2.     What is CIOA’ in line 202 short for? replace in line 252.

3.     I don’t think figure 1 in line 140 is related to the above context.

4.     Some texts in the Figure 1 cannot be distinguished, which requires higher resolution.

5.     Explain what the ‘red lightning’ in Figure 2 represents?

6.     The following recent publication may be relevant to your manuscript and may be used in Table 1.

Sun, Z., Liu, Q., Lv, Z., Li, J., Xu, X., Sun, H., ... & Shi, D. (2022). Targeting macrophagic SHP2 for ameliorating osteoarthritis via TLR signaling. Acta Pharmaceutica Sinica B.

7.     References need to be revised according to ‘Instructions for Authors’.

Reviewer 2 Report

This review article about the therapeutic potential of macrophage polarization by mesenchymal stem cells and paracrine mediators in osteoarthritis describes in detail the current knowledge in this field. The main mechanisms of macrophage interaction with resident cells are reported and provide background knowledge. Mesenchymal stem cells and the secretome as mediator for macrophage polarization are highlighted. However the reviewer advises minor changes:

Line 30 Figure 1: Please provide a more detailed figure legend to describe the major effects

Figure 2: Well described mechanisms.

Line 153: Please explain what you mean with "100-1000 times powerful...on molecular basis"

Table 1: well written

LL.283-300: Please provide a clear definition of the term "MSC" (molecular profile, plastic adherent, differentiation potential and discuss that this defintion is based on in-vitro findings.

L- 316: Please provide a reference

Ll. 479-500: Please provide a clear definition of "extracellular vesicles" and reference the MISEV criteria (PMID: 30637094)

Table 2: this table does not show any OA/ chondrocyte inflammation models. Please change or delete the table. E.g. the references PMID 35057811 and PMID: 33350983 could be of interest

Priming enhances mesenchymal stem cell immunomodulation: well written. Please also discuss and highlight that preconditioning can also lead to adverse effects.

Future perspective and Conlusions: well written
